# Optimal Monitoring Technology for Pediatric Thyroidectomy

**DOI:** 10.3390/cancers14112586

**Published:** 2022-05-24

**Authors:** Daqi Zhang, Hui Sun, Hoon Yub Kim, Antonella Pino, Serena Patroniti, Francesco Frattini, Pietro Impellizzeri, Carmelo Romeo, Gregory William Randolph, Che-Wei Wu, Gianlorenzo Dionigi, Fausto Fama’

**Affiliations:** 1Division of Thyroid Surgery, China-Japan Union Hospital of Jilin University, Jilin Provincial Key Laboratory of Surgical Translational Medicine, Changchun 130033, China; zhangdq@jlu.edu.cn; 2Department of Surgery, KUMC Thyroid Center, Korea University Hospital, Korea University College of Medicine, Seoul 02841, Korea; hoonyubkim@korea.ac.kr; 3Department of Human Pathology of Adulthood and Childhood, University of Messina, 98125 Messina, Italy; pino.antonella@virgilio.it (A.P.); serenapatroniti@gmail.com (S.P.); pietro.impellizzeri@unime.it (P.I.); romeo.carmelo@unime.it (C.R.); 4Division of Surgery, Istituto Auxologico Italiano IRCCS (Istituto di Ricovero e Cura a Carattere Scientifico), 20122 Milan, Italy; francescofrattini79@gmail.com (F.F.); gianlorenzo.dionigi@unimi.it (G.D.); 5Division of Thyroid and Parathyroid Endocrine Surgery, Department of Otolaryngology-Head and Neck Surgery, Massachusetts Eye and Ear Infirmary, Harvard Medical School, Boston, MA 02115, USA; gregory_randolph@meei.harvard.edu; 6Department of Surgery, Massachusetts General Hospital, Harvard Medical School, Boston, MA 02115, USA; 7Department of Otorhinolaryngology, Kaohsiung Medical University Hospital, Kaohsiung Medical University, Kaohsiung 80756, Taiwan; cwwu@kmu.edu.tw; 8Department of Pathophysiology and Transplantation, University of Milan, 20122 Milan, Italy

**Keywords:** thyroid surgery, recurrent laryngeal nerve, intraoperative neural monitoring, children, pediatric patients, endotracheal tube electrode, surgical technique, transcartilage recording electrodes

## Abstract

**Simple Summary:**

Pediatric thyroid surgery is an increasing treatment option for children with thyroid cancer, thyroid nodules, or other thyroid diseases unresponsive to medical treatments. Intraoperative neural monitoring in adults as well as in children proves to be very helpful in preserving the functionality of the recurrent laryngeal nerve. This neural monitoring, which can be carried out with intermittent or continuous (automatic and periodic) stimulation, is of considerable importance in patients aged under 18 years, who are more likely to suffer injuries due to the anatomical features of the recurrent laryngeal nerve itself, which is harder to identify since it is thinner.

**Abstract:**

This retrospective study aimed to describe, firstly, characteristics and outcomes of the intraoperative neural monitoring technology in the pediatric population, and secondarily the recurrent laryngeal nerve complication rate. Thirty-seven patients (age <18 years) operated on from 2015 to 2021 by conventional open thyroid surgery were included. Twenty-four (64.9%) total thyroidectomies and 13 (35.1%) lobectomies were performed. Seven central and six lateral lymph node dissections completed 13 bilateral procedures. Histology showed malignancy in 45.9% of the cases. The differences between the electromyographic profiles of endotracheal tubes or electrodes for continuous monitoring were not statistically significant. In our series of young patients, both adhesive (even in 4- or 5-year-olds) and embedded endotracheal tubes were used, while in patients 3 years old or younger, the use of a more invasive detection method with transcartilage placement recording electrodes was required. Overall, out of 61 total at-risk nerves, 5 (8.2%) recurrent laryngeal nerves were injured with consequent intraoperative loss of the signal; however, all these lesions were transient, restoring their normal functionality within 4 months from surgical procedure. To our knowledge, this is the first study of intraoperative neural monitoring management in a cohort of Italian pediatric patients.

## 1. Introduction

Pediatric thyroid surgery is an increasing treatment option for children with thyroid cancer, thyroid nodules, or with thyroid disease unresponsive to other treatments [1,2].

Thyroidectomy is challenging not only because of the thyroid pathology to be operated on but also because of the anatomic characteristics of young children [3,4,5,6,7,8,9,10,11,12,13,14,15]. Children are more likely to have recurrent laryngeal nerve (RLN) injury than adults [2]. The surgical complexity is mainly related to the difficulty in identifying the RLN, which is smaller than in adults and may be obscured by ectopic cervical thymus tissue [3,4,7]. Collateral RLN fibers innervate the thymus [7]. The baby’s larynx and trachea are smaller than the adult larynx [11,12,13,15]. The overall laryngeal structure is softer in infants than in adults, and although less susceptible to blunt trauma, it is more prone to collapse due to the inspiratory negative pressure created during breathing [14,15] (Table 1).

It is important to refer the young patient to a surgeon experienced in thyroid surgery [1,2]. In addition, the use of a magnifying binocular device and intraoperative neural monitoring (IONM) allows a more reliable identification of pediatric neck structures [16,17,18,19,20].

To date, there are limited data on IONM in pediatric surgery. The primary purpose is to present our clinical experience and technical comments on the use of continuous IONM (C-IONM) in pediatric thyroid surgery. The secondary one is to evaluate the results on the injury rate of RLN.

## 2. Materials and Methods

### 2.1. Study Design

Retrospective cohort study of pediatric patients undergoing thyroid surgery. Data were collected from a prospective clinical medical record database. The database is maintained with quality assurance by an informatics specialist.

### 2.2. Environment

Level 3 College Hospital.

### 2.3. Ethics

This retrospective analysis on pediatric patients undergoing thyroid surgery was approved by the internal review committee (Istituto Auxologico Italiano IRCCS, European Endocrine Surgical Registry, EUROCRINE 2022_01_25_05). Both parents or legal guardians signed informed consent for prospective data collection, surgery, and the IONM procedure before surgery [21]. The details of the specific IONM informed consent were recently published [21]. Participants were assured of the state of anonymity. The study complies with the Declaration of Helsinki of the World Medical Association [22].

### 2.4. Eligibility Criteria for Young Participants

The study included pediatric patients (age <18 years) undergoing transcervical thyroid surgery for the first time. Surgery was performed by a single endocrine IONM-experienced surgeon (G.D.). Surgical procedures which did not involve the thyroid gland and the RLN monitoring (i.e., tracheoesophageal fistula, tracheobronchomalacia, esophageal atresia, and cardiac surgery) were excluded. Revision of case series of IONM surgery not involving the thyroid gland falls out of this report. A recent review of IONM in pediatric cardiac and thoracic surgery is available [23]. Withdrawal of informed consent by parents or legal guardians is a criterion for modification of assigned procedures. Endoscopic procedures were excluded from this analysis.

### 2.5. Schedule of Participants

The total duration of the review was 2015–2021.

### 2.6. Interventions

Our surgical center uses two monitoring systems (Nerve Integrity Monitor (NIM) Response 3.0 System, recently NIM Vital (Medtronic Xomed, Jacksonville, FL, USA) and C2 Nerve Monitoring (Inomed Medizintechnik GmbH, Emmendingen, Germany).

#### 2.6.1. Intermittent IONM Technology

RLN is intermittently stimulated with a sterile, single-use, pulsatile monopolar stimulator probe. The stimulation level is set at 30 Hz, four stimulations per second, a stimulation duration of 100 ms, and an impedance of 5 kΩ [24].

We routinely set the probe so that it could deliver an electric current of 2 mA of intensity for identification of both the vagus nerve (VN) and the RLN, whereas an intensity of 1–0.5 mA was used for RLN and confirmation of the branch [24,25].

Standard IONM procedures were followed and electromyography (EMG) signals with the largest amplitude (V1-R1-R2-V2, vagal and recurrent pre- and post-dissection values, respectively) were recorded [24].

#### 2.6.2. Continuous IONM Technology

C-IONM consisted of the use of sterilized disposable accessories for automatic periodic stimulation (APS, Medtronic, Jacksonville, FL, USA) or the Delta stimulation electrode (Inomed Medizintechnik GmbH, Emmendingen, Germany) (Figure 1a,b). The Delta electrode hooks the VN and is one size only. The APS electrode encircles the VN and there are two APS sizes (2 and 3 mm).

After opening the thyroid sheath through a 2 cm pouch, both C-IONM electrodes were carefully placed on the VN. A careful circumferential VN dissection is required [24,25,26,27]. After having connected the C-IONM electrode to the monitoring system, baselines for latency and amplitude of the evoked response are being automatically calibrated to serve as control data. An upper limit for latency (+10%) and a lower limit for amplitude (−50%) are also set as separate alarm lines. Removal of the C-IONM electrode is performed gently.

#### 2.6.3. EMG Endotracheal Tube (ETT) Selection

An appropriate EMG ETT is generally selected for young patients based on their age, body weight, and size (Figure 2a,b). An uncuffed ETT with an internal diameter (ID) of 3.5 mm is selected for infants up to 1 year of age. A cuffed ETT with an ID of 3.0 mm may be used for infants more than 3.5 kg. and <1 year of age. For 2-year-old patients or older, the age parameter is rounded to provide an EET size that is likely to pass through the vocal cords and have a tight enough seal for ventilation (uncuffed ETT) or to allow for ventilation after cuff inflation (cuffed ETT). Smallest Medtronic NIM TriVantage EMG ETTs available is cuffed with a 5.0 mm inner diameter and a 6.5 mm outer diameter, appropriate for a child aged 4 years and older. Inomed (Emmendingen, Germany) provides adhesive electrodes that rest on ETTs. The adhesive electrode is wrapped around the distal ETT, and the patient is intubated with the electrode positioned at the vocal folds (VF) [26,27,28,29,30,31]. Inomed stikers have 8 contacts, which generate 4 channels. You can overlap them, or if the tube is too small, the last 2 contacts, i.e., the silver wires, can be cut. Thereafter, the adhesive surface electrodes function similarly to the integrated surface electrodes with the same inherent benefits and challenges. Multiple channel electrodes provide the surgeon with the laterality of nerve stimulation.

#### 2.6.4. Transcartilage (TC) Recording Electrodes

In procedures performed using TC recording electrodes, paired 12 mm standard needle electrodes are gently inserted at an oblique angle into each side of the subperichondrium of the lateral surface of the thyroid alar cartilage (Figure 3). This technique has been described in detail [18,32].

#### 2.6.5. Troubleshooting Algorithm

Standard procedures for loss of signal (LOS) troubleshooting were according to the International Nerve Monitoring Study Group (INMSG) guidelines [24].

### 2.7. Primary Objective

The aim of this retrospective study was to describe the technical characteristics and outcomes of IONM technology in the pediatric population. It notably deals with: description of technology used, stratified by age, gender and weight of children, type of EMG ETT used (adhesive or embedded electrodes), tube size ID, type of C-IONM electrodes, selected APS sizes, IONM adverse events (Table 2 describes adverse event reporting), EMG profiles, and troubleshooting algorithm application [24].

### 2.8. Secondary Outcome

The following RLN morbidities were evaluated: transient or permanent laryngeal nerve injury, unilateral or bilateral. RLN paralysis rates were calculated for at-risk nerves (NAR). Preoperative (L1) and postoperative (L2) follow-up included a review of VF mobility by means of a pediatric laryngoscopy 24–48 hours before and at 1–2-day intervals after surgery by an independent laryngologist [31]. Any restriction of VF movement was recorded as postoperative VF paralysis. Patients with documented postoperative VF paralysis underwent repeat examinations at regular intervals 3 and 6 months after surgery until full functional recovery was usually confirmed [31]. Postoperative hypocalcemia was defined when, in at least two measurements, ionized calcium was less than 4.2 mg/dL (normal range in children: 4.8 to 5.3 mg/dL). Hypocalcemia was considered permanent if a patient with hypocalcemia still needed postoperative calcium supplementation beyond 12 months.

### 2.9. Statistical Analysis

Sample size was not measured. For continuous data, the mean values (±standard deviation, range) were calculated, and a paired *t*-test was used for comparison before (start) and after (end) training. A *p* value < 0.05 was considered statistically significant. Data were analyzed using the software program STATA 12.0 (StataCorp LLC, College Station, TX, USA), https://www.stata.com/ (accessed on 17 April 2022).

## 3. Results

### 3.1. Demographic Data

During the 6-year study period, 37 pediatric patients underwent surgery. The mean age was 13.7 years (±2.9, 1–17). Twenty (54.1%) girls and 17 (45.9%) boys. Mean body mass index (BMI) was 19.8 (±4.1, 17.4–29.3). Complete follow-up was available for all patients.

### 3.2. Procedures

On the whole, 24 (64.9%) bilateral procedures (of which 11 total thyroidectomies (TT), 7 TT with central neck dissection, and 6 TT with lateral dissection) and 13 (35.1%) lobectomies were performed. The mean duration of surgery was 79.3 (±5.1, 40–149) minutes. The parathyroid glands were re-implanted in seven (18.9%) patients. All surgical procedures were performed by a surgeon (G.D.) with over 15 years of experience in endocrine surgery and IONM.

### 3.3. Pathology

Final diagnoses showed benign disease in 20 (54.1%) cases and malignancy in the remaining 17 (45.9%); among the latter, 10 patients (27.0%) had lymph node metastases. Mean nodal size was 12.1 (±5.6, 7–24) mm.

### 3.4. Morbidity

No mortality occurred. Postoperative complications were: one (2.7%) cervical hematoma requiring reoperation, five (13.5%) transient hypocalcemia, one (2.7%) permanent hypocalcemia, and one (2.7%) wound seroma.

### 3.5. IONM and RLN Outcomes

#### 3.5.1. NAR

The number of NAR was 61. Five (13.5%) children did not perform L1 and L2 follow-up. In 37 (60.7%) NARs, the RLNs were correct, and non-RLN or infiltrated RLN was found in this series.

#### 3.5.2. EMG Profile

Overall, analysis of the IONM data revealed mean V1 amplitude values of 1163.4 (±417.8, 652–2181) µV on the right side and 1094.1 (±377.3, 590–1983) µV on the left side, and V2 998.9 (±589.7, 157–1430) µV on the right, and 666.5 (±309.2, 220–1149) µV on the left side, respectively.

#### 3.5.3. RLN Injury

Five (8.2%) NARs showed intraoperative LOS. All RLN injuries were transient and regressed within 3.5 months on average. One boy (2.7%, 17 years old, Graves’ disease) underwent staged surgery because he had LOS on the first side of the resection. Complete thyroidectomy was performed 4 months later.

#### 3.5.4. IONM Technology

Nineteen (51.4%) Delta C-IONM probes and eighteen (49.6%) APS (13 (72.2%) 2 mm and 5 (27.8%) 3 mm) were used. There was no morbidity associated with the positioning of the C-IONM electrodes. Figure 4 shows the stratification of IONM technology used according to age of the pediatric patient. Graphs (not shown) are similar for weight, height, and BMI. TC electrodes achieved significantly higher EMG amplitudes. Mean pre-section EMG amplitudes recorded from TC and ETT electrodes were 2058 ± 558 µV vs 814 ± 358 µV (*p* < 0.001) for V1 signal. No statistically significant differences either between the distinct ETTs or between the other types of C-IONM electrodes for EMG profiles were found.

#### 3.5.5. Troubleshooting Algorithm

No cases of TC electrodes dislocation were observed. Dislocation of the C-IONM probe occurred in one case (2.7%, 3 mm APS). ETT dislocation occurred in six (16.2%) patients (three stickers and three preformed ETT). Dislocation occurred in children with a mean age of 7 years, all aged less than 10. No patients older than 11 years experienced ETT dislocation.

## 4. Discussion

Thyroid disease, a particularly malignant disease, is rare in children. Thyroid surgery in pediatric patients differs from traditional surgery performed on adults for the marked differences of both anatomical and physiological order, i.e., the diameters of the upper airways and the dimensions of the vascular, glandular, and nerve structures, which require an even more careful and delicate surgical dissection. Furthermore, the presence of other anatomical structures, such as the thymus, have not yet gone into physiological involution can add further difficulties to the execution of the surgical procedure [16]. The use of IONM in pediatric surgery presents challenges, including adapting equipment to young patients [16,17,18,19,20,32]. The INMSG has recently published a comprehensive guideline [18]. The benefits of IONM, already known for thyroid surgery in adults, were also highlighted in this study. The use of the C-IONM (both with Delta and APS probes) has made it possible to minimize permanent nerve injuries even in children. In fact, only five nerve injuries have been transient, with a total regression within 4 postoperative months. To our knowledge, this is the first study of IONM management in a cohort of Italian pediatric patients.

### 4.1. ETT & TC

The smallest Medtronic available EMG ETT has an inner diameter of 5.0 mm and an outer diameter of 6.5 mm, which makes it suitable for children 4 years old or older. Therefore, ETTs are offered with stickers or TC electrodes for younger children. Regardless of whether an ETT is used with or without a cuff, additional tubes half a size larger and half a size smaller should be available for endotracheal intubation attempts [18,24]. In our series, self-adhesive surface electrodes on ETT have been used successfully in children older than 4 years of age. TC electrodes have been used successfully in children as young as 1 year of age, whereas ETT dislocation occurred in 16.2% of patients. Confirmation of correct ETT electrode placement in children is difficult due to the small size of the larynx. 

For all these reasons, we emphasize the importance of IONM curricula for both surgeons and anesthesiologists [18,32].

### 4.2. C-IONM

The C-IONM probe requires that the electrode and VN be of similar size to allow stable stimulation. This may prevent the use of 3 mm APS systems in very young children (only 5 APS of 3 mm have been used in our experience). The 3 mm ones are too large for the small VN, with little contact and the probe tending to slide on the VN itself. There was no morbidity associated with the positioning of the C-IONM electrodes. Nevertheless, we recommend a careful neural monitoring and testing of all possible consequences of VN electrode placement and VN stimulation in small patients.

### 4.3. EMG Profiles

In this series of pediatric patients, high V1 amplitude values (average 1163.4 µV on the right side and 1094.1 µV on the left side) were observed.

### 4.4. RLN Morbidity

The 8.2% of NAR showed intraoperative LOS. All RLN lesions were transient and regressed within 3.5 months on average. Schneider R. et al. performed a retrospective electrophysiological study of 504 patients from 1998 to 2016, including all patients aged 18 years or younger who underwent thyroidectomy with RLN monitoring. The author concluded that continuous IONM is more accurate than intermittent IONM in pediatric thyroidectomies [20].

### 4.5. Limitations

This is a retrospective study, whose limited number of patients participating is too small to perform a statistical analysis of RLN results. There is no comparison with a control group. We kindly suggest you to be cautious in reproducing our results.

## 5. Conclusions

In our study, the optimal technology for neural monitoring in children was stratified based on the patient’s age. We hope that future technological improvements will favor and simplify this surgery as well, by adapting the surgical instruments to the anatomical characteristics of pediatric patients.

Being a highly specialized surgery, it is preferable to refer these patients to specialized centers and with other volumes in this type of surgical procedure. Further studies with higher patient volumes or multicenter studies will be able to support what is described in our experience.

## Figures and Tables

**Figure 1 cancers-14-02586-f001:**
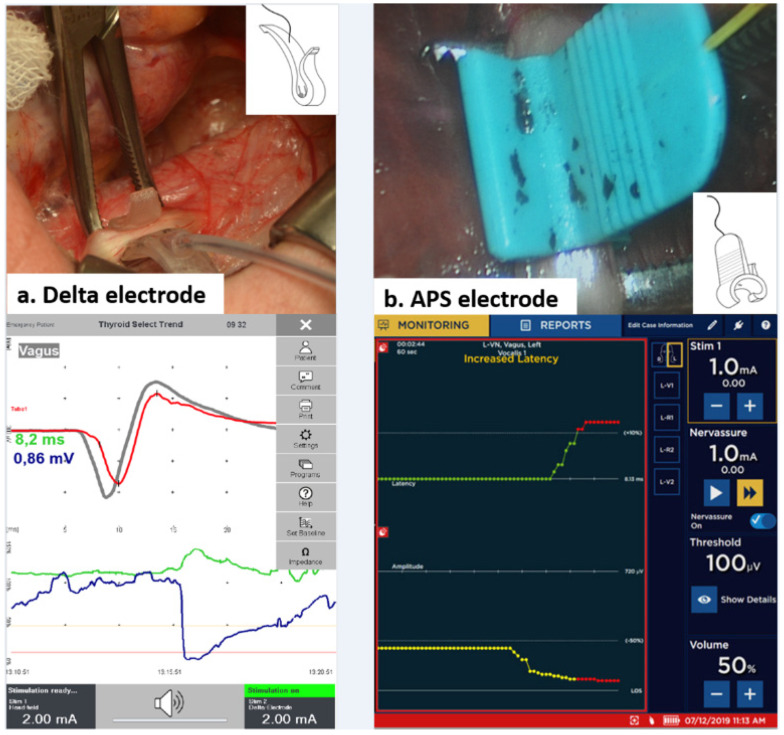
Representation of different C-IONM electrodes (geometry and position of probe on the VN): (**a**) Delta and (**b**) APS, with their relative EMG profiles obtained below. In both cases, a 360° dissection of the VN is required. The VN of babies is small, and the APS most frequently used in children was 2 mm. Abbreviations: VN: vagus nerve; C-IONM: continuous intraoperative neural monitoring; APS: automatic periodic stimulation; EMG: electromyography.

**Figure 2 cancers-14-02586-f002:**
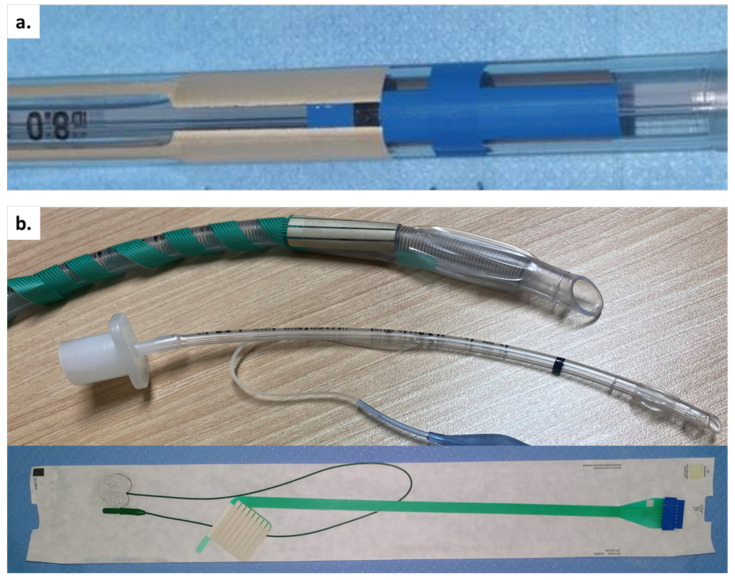
(**a**) Embedded and (**b**) adhesives EMG ETT. It is difficult to place the stickers on tubes with an ID below 4 mm. In order not to overlap the electrodes/stickers, you can cut the last 2 rows of wires. Abbreviations: ID: internal diameter; ETT: endotracheal tube; EMG: electromyography; TC: transcartilage recording electrode.

**Figure 3 cancers-14-02586-f003:**
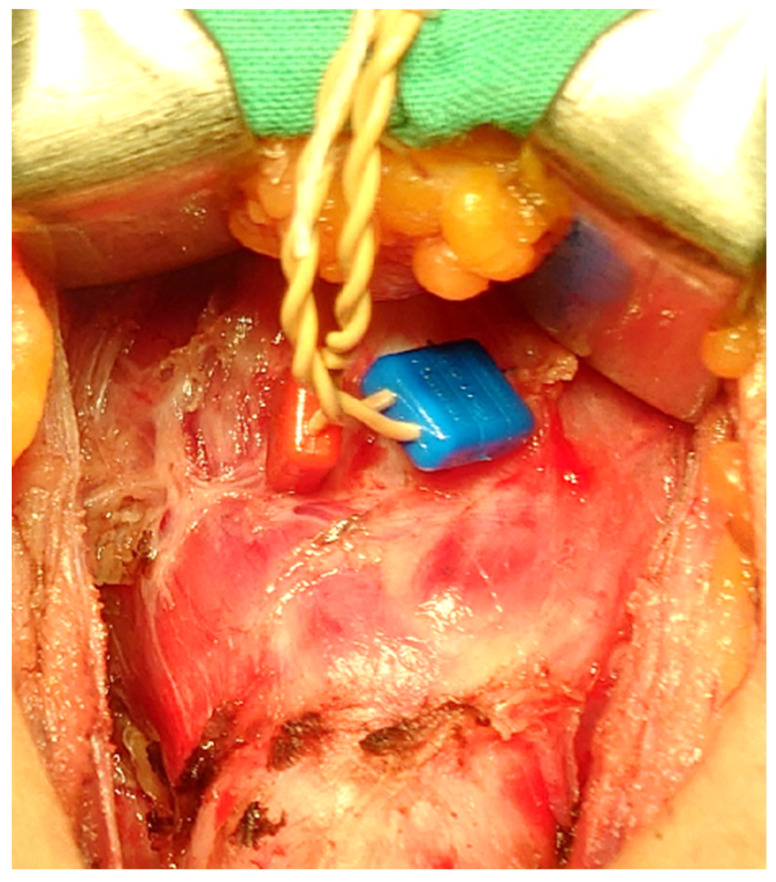
TC recording electrodes through the thyroid cartilage and perichondrium. Ideal but invasive technique when it is not possible to use the electrodes placed on ETT. The placement of the TC must be meticulous because the children’s larynx is small and soft. Abbreviations: TC: transcartilage recording electrode; ETT: endotracheal tube.

**Figure 4 cancers-14-02586-f004:**
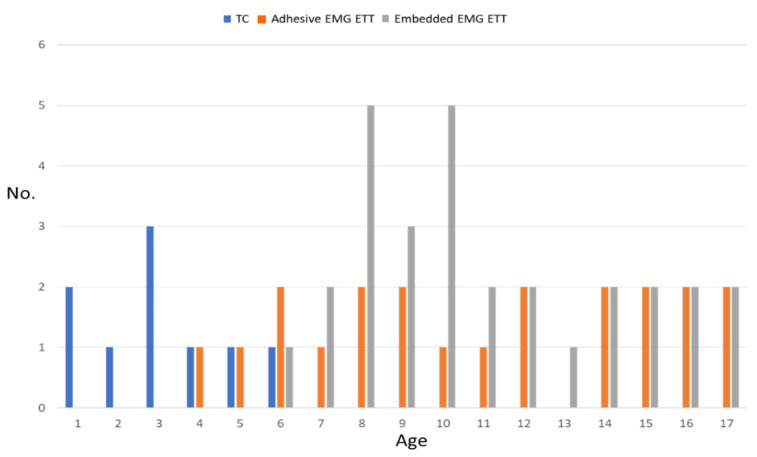
Stratification of IONM technology used according to age. Abbreviations: No: number; IONM: intraoperative neural monitoring; TC: transcartilage recording electrode; EMG: electromyography; ETT: endotracheal tube.

**Table 1 cancers-14-02586-t001:** Peculiarities of the neck in children. Knowledge about these variations is extremely important in clinical aspects like intubation, laryngoscopy, radiological images, thyroidectomy, and parathyroidectomy.

Anatomical Peculiarities of the Neck in Children
Small thyroid gland volume
Thin RLN
Thin RLN branches
Thin EBSLN
Laryngo-tracheomalacia (softer cartilaginous framework of trachea and larynx)
Narrow larynx and trachea
Larynx is more anterior
At the glottic (vocal fold) level, the larynx is approximately one third the adult size
Larynx is situated higher in the neck
Angle between the epiglottis and vocal cords is more acute in the infant, thus making direct visualization more difficult
Small parathyroid glands
Small thyroid arteries and veins
Hypertrophic thymus
Thymus superimposed on the thyroid gland
Collateral RLN fibers innervate the thymus
Possible congenital anomalies

Abbreviations: RLN: recurrent laryngeal nerve; EBSLN: external branch superior laryngeal nerve.

**Table 2 cancers-14-02586-t002:** Adverse event reporting.

Concern	Adverse Event
EMG tube	Displacement of adhesive surface electrodes Displacement of the endotracheal tube Tube change with larger/smaller ID tube Laryngeal/vocal cord injury ^
Carotid sheet pocket creation & C-IONM electrode implantation	VN injury * Vocal cord palsy * Vascular injury (carotid artery) Vascular injury (jugular vein)
VN stimulation	Bradycardia
C-IONM electrode displacement/replacement	
Hematoma	
Infection	
Allergies	
Surgical IONM break/malfunction	
Surgical C-IONM fracture/malfunction	

Abbreviations: VN: vagus nerve; ID: internal diameter; EMG: electromyography; IONM: intraoperative neural monitoring; C-IONM: continuous intraoperative neural monitoring. ^ postoperative follow-up included VF check performed via pediatric laryngoscopy in a range of 1–2 days by an independent laryngologist * during VN dissection and after the C-IONM electrode was placed, the VN was stimulated repeatedly by means of the intermitted stimulating probe, proximally and distally to the location of automatic periodic stimulation (APS), to verify whether the dissection or electrode placement determined VN injury.

## Data Availability

All data were collected from a prospective clinical medical record database. The database is maintained with quality assurance by an informatics specialist.

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
