# Peer review of "Optimal Monitoring Technology for Pediatric Thyroidectomy"

_cancers, 2022, doi:10.3390/cancers14112586_

Round 1

Reviewer 1 Report

The article addresses an interesting topic for surgeons specializing in thyroid surgery.

The sample is small in a single hospital setting to obtain applicable practical conclusions.

It would be interesting to carry out multicenter research with a greater number of patients and stratify the results by age.

Author Response

We thank the Reviewers for their comments and constructive criticism.

We have performed all suggested modifications in a point-by-point manner; changes are typed in red text. As a result, we believe that the quality of our paper has been significantly improved.

Reviewer #1:

We thank Reviewer # 1 for his/her comments.

R: We are aware that the cohort of patients is not very large, and we appreciate the suggestion of an expansion of the case series with the involvement of other centers for future further investigations in the field. Pediatric thyroidectomy is rare, and very specialized, and to make research on larger volumes it is necessary to do a multicenter study. We believe that the role of scientific societies is important in proposing this study.

Reviewer 2 Report

The authors present their experiences with IONM in children. The study has potential, but some important changes have to be made before considering it for publication.

Minor comments:

Line 71 - Table 1: this table is difficult to read

Line 81 – were all the patients from a single center? (since the authors seem to be from many centers around the world, maybe it would be worth to mention where the patients were from?)

Line 178 – data in the table misplaced

Line 195 - calcium under 4.2 mg/dL – please add reference range

The results section: Perhaps a table summarizing the results would be easier and faster to read?

Line 294 - To our knowledge, this is the first study of IONM management in a cohort of Italian pediatric patients – please move this statement to the discussion section.

Line 291 - There is no comparison with a control group. – what kind of control group would be suitable for this study?

Crucial comments:

Line 296 - In our study, the optimal technology for neural monitoring IN CHILDREN was stratified based on the patient’s age

  1. add “in children” into this sentence
  2. If this is the conclusion of the work, in my opinion the title of the work should be changed to something like “The best intra-operational RLN monitoring technique in children” or “How to chose the best technique for ….”

Line 297 - these results are related to some technological features of IONM – this sentence should be changed into something much more precise and specific. Instead of “these results” write exactly what results you are describing. Instead of “some technological features” try to list the features and their association to particular monitoring modalities. Change this sentence into the core point of the conclusion section.

Additional comment: seems that the authors wanted include some non-published material, but there’s somebody’s CV in that section :)

Author Response

We thank the Reviewers for their comments and constructive criticism.

We have performed all suggested modifications in a point-by-point manner; changes are typed in red text. As a result, we believe that the quality of our paper has been significantly improved.

Reviewer #2:

The authors present their experiences with IONM in children. The study has potential, but some important changes have to be made before considering it for publication.

We thank Reviewer # 2 for having appreciated our work.

Minor comments:

Line 71 - Table 1: this table is difficult to read

R: This table lists the peculiarities of the neck in children that can influence preoperative diagnostics, intubation and the surgical procedures themselves. We have reformatted the presentation of this table.

Line 81 – were all the patients from a single center? (since the authors seem to be from many centers around the world, maybe it would be worth to mention where the patients were from?)

R: All patients were Italian and were surgically treated by G.D. Author first at the University of Insubria (Varese), then at the University of Messina, and recently at the University of Milan. The foreign co-authors have actively collaborated, with their extensive experience in thyroid surgery, intraoperative neural monitoring (IONM), and pediatric procedures, in the processing of the data and the drafting of the manuscript.

Line 178 – data in the table misplaced

R: This table shows the potential concern (in the left column) and the related adverse events (in the right column). We have reformatted the presentation of this table.

Line 195 - calcium under 4.2 mg/dL – please add reference range

R: The ionized calcium normal range has been added in the text.

The results section: Perhaps a table summarizing the results would be easier and faster to read?

R: The results are presented schematically with subheadings, therefore adding a table could be redundant. We consider the information entered to be very important. If the auditor insists, however, we can change it.

Line 294 - To our knowledge, this is the first study of IONM management in a cohort of Italian pediatric patients – please move this statement to the discussion section.

R: The sentence, as advised, has been moved into discussion section.

Line 291 - There is no comparison with a control group. – what kind of control group would be suitable for this study?

R: In this retrospective study, we decided to present the data without comparing them with a control group. Thank you for the clarification. The control group may be comparing the intermittent versus continuous monitoring method. A control group without monitoring is not feasible.

Crucial comments:

Line 296 - In our study, the optimal technology for neural monitoring IN CHILDREN was stratified based on the patient’s age –  add “in children” into this sentence

R: As advised, the words “in children” have been added to the sentence.

If this is the conclusion of the work, in my opinion the title of the work should be changed to something like “The best intra-operational RLN monitoring technique in children” or “How to chose the best technique for ….”

R: Thank you for your suggestion. If the reviewer agrees we can modify the title into: “Optimal monitoring technology for pediatric thyroidectomy.”

Line 297 - these results are related to some technological features of IONM – this sentence should be changed into something much more precise and specific. Instead of “these results” write exactly what results you are describing. Instead of “some technological features” try to list the features and their association to particular monitoring modalities. Change this sentence into the core point of the conclusion section.

R: This sentence has been modified.

Reviewer 3 Report

The Authors report, within a retrospective observational study, on the characteristics and outcomes of intraoperative neural monitoring technology in children undergoing thyroid surgery, together with the complication rate on the recurrent laryngeal nerve. The article is of interest. Few comments.

  • Can you please clarify whether this is a multicentric study, given the authorship, or whether the patient population is collected within a single hospital? In this last case, why is the authorship so diverse?
  • Table 1 and 2 are misaligned and hard to read in the present version.
  • Can you point out the Hospital(s) in which the study was carried out?

Author Response

We thank the Reviewers for their comments and constructive criticism.

We have performed all suggested modifications in a point-by-point manner; changes are typed in red text. As a result, we believe that the quality of our paper has been significantly improved.

Reviewer #3:

The Authors report, within a retrospective observational study, on the characteristics and outcomes of intraoperative neural monitoring technology in children undergoing thyroid surgery, together with the complication rate on the recurrent laryngeal nerve. The article is of interest.

We thank Reviewer # 3 for having appreciated our work.

Few comments.

Can you please clarify whether this is a multicentric study, given the authorship, or whether the patient population is collected within a single hospital? In this last case, why is the authorship so diverse?

R: All patients were Italian and were surgically treated by G.D. Author first at the University of Insubria (Varese), then at the University of Messina, and recently at the University of Milan. The foreign co-authors have actively collaborated, with their extensive experience in thyroid surgery, in the processing of the data and the drafting of the manuscript.

Table 1 and 2 are misaligned and hard to read in the present version.

R: Table 1 lists the peculiarities of the neck in children that can influence preoperative diagnostics, intubation and the surgical procedures themselves. Table 2 shows the potential concern (in the left column) and the related adverse events (in the right column). We have reformatted the presentation in both tables.

Can you point out the Hospital(s) in which the study was carried out?

R: as mentioned above, all surgical procedures were performed first at the University of Insubria (Varese), then at the University of Messina, and recently at the University of Milan.

Round 2

Reviewer 2 Report

I have no further commentaries